# The impact of the COVID-19 pandemic on microbial keratitis presentation patterns

Gibran F. Butt[1,2], Alberto Recchioni[1,2], George Moussa[2], James Hodson[3], Graham R. Wallace[1], Philip I. Murray[1,2], Saaeha Rauz[1,2]*

1 Academic Unit of Ophthalmology, Institute of Inflammation and Ageing, University of Birmingham, Birmingham, United Kingdom, 2 Birmingham & Midland Eye Centre, Sandwell and West Birmingham Hospitals NHS Trust, Birmingham, United Kingdom, 3 Queen Elizabeth Hospital Birmingham, University Hospitals Birmingham NHS Foundation Trust, Birmingham, United Kingdom

* s.rauz@bham.ac.uk

**Data Availability Statement:** Data can be found at the following URL: DOI https://doi.org/10.6084/m9.figshare.14873052.

**Funding:** This study was supported by a Medical Research Council Developmental Pathway Funding

## Abstract

### Background

Microbial keratitis (MK) is the most common non-surgical ophthalmic emergency, and can rapidly progress, causing irreversible sight-loss. This study explored whether the COVID-19 (C19) national lockdown impacted upon the clinical presentation and outcomes of MK at a UK tertiary-care centre.

### Methods

Medical records were retrospectively reviewed for all patients with presumed MK requiring corneal scrapes, presenting between 23rd March and 30th June in 2020 (Y2020), and the equivalent time windows in 2017, 2018 and 2019 (pre-C19).

### Results

In total, 181 and 49 patients presented during the pre-C19 and Y2020 periods, respectively. In Y2020, concurrent ocular trauma (16.3% vs. 5.5%, p = 0.030) and immunosuppression use (12.2% vs 1.7%, p = 0.004) were more prevalent. Despite proportionately fewer ward admissions during the pandemic (8.2% vs 32.6%, p<0.001), no differences were observed in baseline demographics; presenting visual acuity (VA; median 0.6 vs 0.6 LogMAR, p = 0.785); ulcer area (4.0 vs 3.0mm$^2$, p = 0.520); or final VA (0.30 vs 0.30 LogMAR, p = 0.990). Whilst the overall rates of culture positivity were similar in Y2020 and pre-C19 (49.0% vs. 54.7%, p = 0.520), there were differences in the cultures isolated, with a lower rate of poly-microbial cultures in Y2020 (8.3% vs. 31.3%, p = 0.022).

### Conclusions

Patient characteristics, MK severity and final visual outcomes did not appear to be affected in the first UK lockdown, despite fewer patients being admitted for care. Concurrent trauma and systemic immunosuppression use were greater than in previous years. The difference in spectra of isolated organisms may relate to behavioural changes, such as increased hand hygiene.

Scheme "Development of a synthetic biomembrane dressing that prevents corneal scarring" (MR/N019016/1). SR, GRW and GFB are funded by the MRC MR/N019016/1. The funders had no role in study design, data collection and analysis, decision to publish, or preparation of the manuscript.

**Competing interests:** The authors have declared that no competing interests exist.

## Introduction

During the UK's first national lockdown in 2020 due to the COVID-19 pandemic, the official parliamentary advice in the UK, and of bodies such as the *Academy of Medical Royal Colleges* [1] urged the public to seek timely medical attention, where required, to avoid unnecessary delay and complications. However, Emergency Departments (EDs) across England experienced a significant decrease in activity during the national lockdown period [2], raising concerns about whether patients were seeking help appropriately. Concurrently, rising numbers of reports describe public apprehensions about attending hospital or seeking medical attention [3, 4]. The pandemic impacted non-COVID patients in manifold ways, for example, with reduced patient attendances for emergency and urgent services, and disruptions to cancer services [5–7]. This has raised concerns globally about the need to mitigate these effects.

Ophthalmology departments provide a highly specialised emergency service, and were amongst the most disrupted by the pandemic. During the pandemic, the reported emergency eye care workload increased in severity and complexity, as did the demand for emergency surgical procedures, despite an overall reduction in attendances [8–10]. The public's apprehensions about engaging clinical services during the pandemic may potentially be causing delays in presentation, diagnosis, and implementation of appropriate management. Poyser *et al. (2020)*, for example, reported fewer cases of retinal tears, but increased cases of macular-off retinal detachments [11]. Similarly, Babu et al. (2020) described the inability to meet the demand for emergency corneal transplants for patients presenting with perforated corneal ulcers [12], also highlighting the secondary impact of the pandemic on supporting services, such as organ donation and retrieval [13].

The Birmingham and Midland Eye Centre (BMEC) is a tertiary referral unit providing seven-days-a-week Emergency and in-patient eye services for the West Midlands, UK. The BMEC ED records approximately 120 attendances per day, with over 450 emergency admissions annually (S1 Table); almost 25% of these are for microbial keratitis (MK) [14], requiring intensive eye drops through the night, as well as daily reviews. MK constitutes the most common non-surgical emergency and reason for admission in eye care services in the UK [15]; it is also rapidly progressive, and requires prompt management to prevent irreversible blindness. Typically, patients with MK would be managed by hospital services after presenting, either by self-referral or by referral from community-based eye care services. Such services are typically delivered by community opticians and general practitioners. However, during the pandemic, such community healthcare services were severely impaired, with more patients instead utilising alternate pathways, such as the NHS non-emergency phoneline 111 [16].

Our group previously characterised public perceptions of eye symptom severity and consequent health seeking behaviour [17]. The comparison of various clinical scenarios in normal and pandemic contexts revealed that respondents felt materially less of an impetus to seek in-person clinical help during the pandemic. For mild self-limiting disease, this study highlighted the potentially beneficial adaptive behaviours that may reduce the risk of COVID-19 exposure; however, this difference was also noted in response to serious conditions such as MK, raising concerns about potential delays in presentation and consequent poorer clinical outcomes. To investigate this further, we explored whether the first UK national lockdown during the SARS-CoV-2 pandemic impacted upon the clinical presentation, causative organisms, admission rates or outcomes of MK. We hypothesized that the presentation and outcomes of patients with MK would be affected by the first UK national lockdown in 2020, compared to previous years.

## Methods

This retrospective study of medical records was conducted in accordance with the Declaration of Helsinki, and in accordance with local institutional policy. All data were anonymised prior to analysis, and the need for consent was waived. Approval was obtained from Sandwell and West Birmingham Hospitals NHS Trust Department of Clinical Effectiveness (registration #1512) to undertake this project as a service evaluation.

### Study design & population

All cases of MK requiring corneal scrapes, presenting between the 23rd March and 30th June 2020 (Y2020) were identified through the regional microbiology service (Black Country Pathology Services Supporting Sandwell and West Birmingham Hospitals NHS Trust) database, and cross-checked with BMEC ED electronic medical records. Patients presenting during the equivalent time windows in the preceding three years (2017, 2018, and 2019) were also identified, and included as the comparator cohort (pre-C19), to reflect the variation of the disease. During the period being studied in 2020, all first-time face to face appointments were replaced with an initial telephone consultation, followed by a face-to-face consultation, where indicated. This had the effect of reducing the overall number of face-to-face appointments [18].

### Routine clinical practice

Clinical assessments and decisions, such as the need for investigations, admission and follow ups, were undertaken by the BMEC ED attending ophthalmologists, in accordance with local guidelines. During the first national lockdown period, the BMEC in-patient ward was closed to allow nursing staff to be redeployed to specified medical wards to undertake general nursing duties, as well as to provide specialist care for admitted ophthalmic patients. Patients requiring admission for urgent care were initially admitted to amber wards (COVID status unknown), before relocating to specified Green (COVID-Free) and Red (COVID positive) wards with designated ophthalmic beds. The decision to admit patients was based on factors including clinical severity, risk of adverse events (e.g., perforation), social care needs (e.g., the ability to diligently administer all drops, proximity to clinic, and ability to attend for daily visits), with the final decision being taken by the lead clinician for any given session.

Where MK was suspected on presentation, corneal scrapes were taken to confirm the diagnosis. The typical corneal sampling kit consisted of a sterile needle (e.g. 23G) or scalpel blade for corneal tissue acquisition, one each of chocolate and blood agar plates, and a Sabouraud's agar plate for fungus. Nutrient depleted agar seeded with Escherichia coli was used for Acanthamoeba cultures, where indicated. Samples were placed on glass slides for microscopy and Gram staining. Dry swabs were also acquired for microbe polymerase chain reaction (PCR) typing. All cultures were incubated according to departmental protocols for at least one week.

A positive isolate was defined as a growth along the line of inoculation on solid media, and poly-microbial keratitis was confirmed if more than one clinically significant organism was isolated. Significant isolates were tested against antibiotics, in accordance with local protocols, using both disc diffusion (the British Society of Antimicrobial Chemotherapy methodology; www.BSAC.org.uk) and Vitek AST systems (www.biomerieux.co.uk). Isolates identified as contaminants in the microbiology reports were excluded from analysis.

### Data collection

All data were recorded in an adaptation of a validated data collection proforma used in a previous study [14], using the secure web application Research Electronic Data Capture (REDCap©

v9.6.3 2020 Vanderbilt University, Nashville, TN, USA). Data collected included patient demographics (sex, age, ethnicity and Index of multiple deprivation [IMD] score) and clinical details (presenting features, underlying risk factors, past ocular history, medications). The IMD score combines information from seven differentially weighted domains, to classify the relative deprivation of small areas around the UK; and scores were obtained from a government website [19].

Underlying risk factors were grouped as follows: contact lenses wear; active ocular surface disease (complete list in S2 Table); previous keratitis (infective and marginal); previous trauma (healed before the onset of MK) or previous ocular surgery; concurrent trauma (related to the onset of MK); foreign bodies associated with the current episode; as well as the systemic conditions: diabetes mellitus; rheumatoid arthritis; thyroid eye disease; and the use of systemic immunosuppression medication.

Details of clinical assessments were also recorded. The best corrected Snellen visual acuity (VA) at presentation was collected, and converted to LogMAR VA for analysis [20]. In addition, the final VA was also recorded, based on assessments performed at clinical follow up appointments (1, 3, 4 and 12 weeks after presentation). Where patients attended a clinical follow up at week 12, the VA at this appointment was used, with the latest available assessment used instead in patients that were discharged from the service prior to this.

Slit-lamp biomicroscopy was used to assess the size of the epithelial defect, infiltrate, or scar, using standardised methodology adapted from the Herpetic Eye Disease Study [21], by measuring the longest and the longest perpendicular dimensions. The area was then calculated by multiplying these readings together. Epithelial defect, infiltrate, and scar size were not differentiated, henceforth this measurement is referred to as the "ulcer area", which was also the summation of all single areas of involvement in the cornea.

The corneal involvement score (CIS) was retrospectively derived from the clinical notes, based upon the validated corneal opacification score described by Ong *et al*. [22]. Briefly, the locations of the corneal ulcer are documented according to the number of quadrants involved (temporal, superior, nasal, inferior), which are each assigned 1 point, with involvement of the central 4mm zone being assigned 5 points. The numbers of points are then added, to give a final CIS out of 9.

## Statistical analysis

Comparisons of patient characteristics by presentation period (Y2020 or pre-C19) were performed, using Fisher's exact tests for nominal variables, and Mann-Whitney U tests for ordinal and continuous variables. Continuous variables were reported as mean ± standard deviation if approximately normally distributed, with median (interquartile range; IQR) used otherwise. Cases with missing data were excluded from the analyses of the affected variables, and the sample sizes included in each analysis are reported in the associated tables. All analyses were performed using IBM SPSS 26 (IBM Corp. Armonk, NY), with $p < 0.05$ deemed to be indicative of statistical significance throughout.

## Results

### Included cases

A total of 230 MK patients were identified, comprising 63, 50, 68 and 49 patients from the time windows in the years 2017, 2018, 2019 and 2020, respectively. Total numbers of attendances to the BMEC for any indication were 12,128 during the time window in 2018 and 12,239 in 2019, compared to only 5,759 in 2020 (accurate data were not available for 2017). As such, MK comprised 0.5% of attendances in 2018–19, which increased significantly to 0.9% in 2020 (p = 0.001). Comparisons between the years 2017–2019 found no significant differences

in patient characteristics (S3 Table). As such, the 181 cases from these three years were combined into a single cohort for subsequent analysis (pre-C19), and compared to the 49 cases from the year 2020 (Y2020).

## Patient characteristics

Comparisons between Y2020 and pre-C19 found no significant differences in the age, sex, laterality of eye, ethnicity or IMD scores between the groups (Table 1). The duration of symptoms at presentation was also similar in the Y2020 and pre-C19 groups, with medians of 4 days (IQR 2–7) and 3 days (1–6), respectively (p = 0.201). Of the risk factors considered, concurrent ocular trauma (16.3% vs. 5.5%, p = 0.030) and systemic immunosuppression (12.2% vs. 1.7%, p = 0.004) were both significantly more prevalent in the Y2020 group. The full list of causes of concurrent trauma is reported in S4 Table.

**Table 1. Characteristics of patients presenting with microbial keratitis.**

| | Pre-C19 | | Y2020 | | |
|---|---|---|---|---|---|
| | *N* | *Statistic* | *N* | *Statistic* | **p-Value** |
| Age (years) | 181 | 55.5 ± 21.1 | 49 | 53.3 ± 17.8 | 0.503 |
| Sex—male (%) | 181 | 95 (52.5%) | 49 | 31 (63.3%) | 0.198 |
| Ethnicity | 164 | | 38 | | 0.864 |
| *White* | | 112 (68.3%) | | 27 (71.1%) | |
| *Asian* | | 38 (23.2%) | | 7 (18.4%) | |
| *Black* | | 9 (5.5%) | | 3 (7.9%) | |
| *Mixed / Other* | | 5 (3.0%) | | 1 (2.6%) | |
| IMD decile | 178 | | 49 | | 0.839* |
| *1–3* | | 96 (53.9%) | | 26 (53.1%) | |
| *4–7* | | 58 (32.6%) | | 19 (38.8%) | |
| *8–10* | | 24 (13.5%) | | 4 (8.2%) | |
| Laterality—right (%) | 181 | 89 (49.2%) | 49 | 27 (55.1%) | 0.521 |
| Duration of symptoms at presentation (days) | 121 | 3 (1–6) | 37 | 4 (2–7) | 0.201 |
| **Risk factors** | | | | | |
| Contact lens | 181 | 60 (33.1%) | 49 | 14 (28.6%) | 0.608 |
| Underlying OSD (active)** | 181 | 80 (44.2%) | 49 | 26 (53.1%) | 0.333 |
| Previous keratitis*** | 181 | 20 (11.0%) | 49 | 7 (14.3%) | 0.616 |
| Previous surgery/trauma | 181 | 31 (17.1%) | 49 | 10 (20.4%) | 0.674 |
| Concurrent trauma | 181 | 10 (5.5%) | 49 | 8 (16.3%) | **0.030** |
| Foreign body | 181 | 3 (1.7%) | 49 | 2 (4.1%) | 0.289 |
| Diabetes mellitus | 181 | 17 (9.4%) | 49 | 7 (14.3%) | 0.304 |
| Rheumatoid arthritis | 181 | 6 (3.3%) | 49 | 1 (2.0%) | 1.000 |
| Systemic immunosuppression | 181 | 3 (1.7%) | 49 | 6 (12.2%) | **0.004** |
| Thyroid eye disease | 181 | 1 (0.6%) | 49 | 0 (0.0%) | 1.000 |

Abbreviations: Pre-C19, Pre-COVID-19 Years (2017,2018,2019); Y2020, year 2020; OSD, ocular surface disease; IMD, index of multiple deprivation; MK, microbial keratitis. Continuous variables are reported as mean ± SD or median (interquartile range), with p-values from Mann-Whitney U tests. Categorical variables are reported as N (column %), with p-values from Fisher's exact tests, unless stated otherwise. Bold p-values are significant at p<0.05. For risk factors, "previous" denotes that the risk factor had healed prior to onset of MK.

*p-Value from Mann-Whitney U test, as the factor is ordinal.

**Ocular surface disease, such as dry eye, affecting the patient at the time of presentation–a full list of included diseases is reported in S2 Table.

***Viral/bacterial/fungal/parasitic/marginal disease.

## Clinical assessments

The severity of disease at presentation was quantified using the VA, ulcer area and CIS, none of which were found to differ significantly between the Y2020 and pre-C19 groups (Fig 1A and Table 2). These comparisons were also repeated after excluding the 67 patients with pre-existing visual impairment at presentation (based on their ophthalmic history), with the difference between groups remaining non-significant.

Despite the similarities in patient characteristics, admission rates were found to be significantly lower in Y2020, at 8.2% compared to 32.6% for the Pre-C19 group (p<0.001, Table 3). However, the disease course was found to be similar in the two groups, with no statistically significant differences noted in the final VA (Fig 1A), or in complication or intervention rates (Table 3).

## Microbiology

Rates of culture positivity were similar in the two groups, at 49.0% in Y2020 and 54.7% in Pre-C19 (p = 0.520, Table 4 and Fig 1B). However, the distribution of culture isolates was found to vary between the groups (Fig 1C), with a significantly lower rate of poly-microbial infections in Y2020, compared to pre-C19 (8.3% vs. 31.3%, p = 0.022), and a non-significant tendency for higher rates of gram-negative mono-microbial infections in Y2020 (33.3% vs. 18.2%, p = 0.160). Fungal infections comprised similar proportions of culture positive cases in both groups (4.2% vs. 5.1% in Y2020 vs. pre-C19, p = 1.000).

Assessment of the most frequent gram-positive isolates found a preponderance of *Staphylococcus aureus* infections in the Y2020 group, being isolated in 25.0% of those with positive cultures, compared to 11.1% in pre-C19 (p = 0.099). Of the gram-negative isolates, it was notable that no cases of *P. aeruginosa* were detected in Y2020, compared to 11.1% in previous years (p = 0.120). However, neither of these differences reached statistical significance, largely as a result of the small sample sizes in these subgroups.

## Discussions

The pandemic's negative impact on ophthalmic services [10, 11, 13, 23, 24] has raised concerns about patients' well-being. This study evaluated the impact of the first COVID-19 lockdown on the outcomes of patients with MK. In a survey completed by the British public, our group identified how concerns about the pandemic would lead individuals to consider seeking healthcare for their eye symptoms less urgently than if there was no pandemic [17]. The present study demonstrates a strong similarity between patients with MK in the first UK lockdown and those from previous years, with respect to time-to-presentation, presenting VA and ulcer area, complications, interventions, and final VA. However, the prevalence of concurrent trauma and use of systemic immunosuppression were greater than in previous years, while fewer poly-microbial infections and ward admissions occurred. Thus, patients presenting to this centre during the lockdown appear to be accessing services on time, did not have worse MK, and perhaps had milder disease in a more vulnerable group of patients.

Disease epidemiology and health care services vary geographically. Whilst Agarwal *et al.* [24] reported an increase in MK incidence during the lockdown at their unit in India, Poyser *at al.* [10] report a decrease in contact lens associated keratitis of more than 50%, compared to the same period in 2019, although the proportions remained similar in both study periods. The present study's results identified an increase in the proportion of MK patients seen in the department compared to previous years, whilst fewer patients were seen in the department overall [18]. The average time-to-presentation and number of patients attending in Y2020 compared to pre-C19 indicate no change in the patterns of the public accessing services for

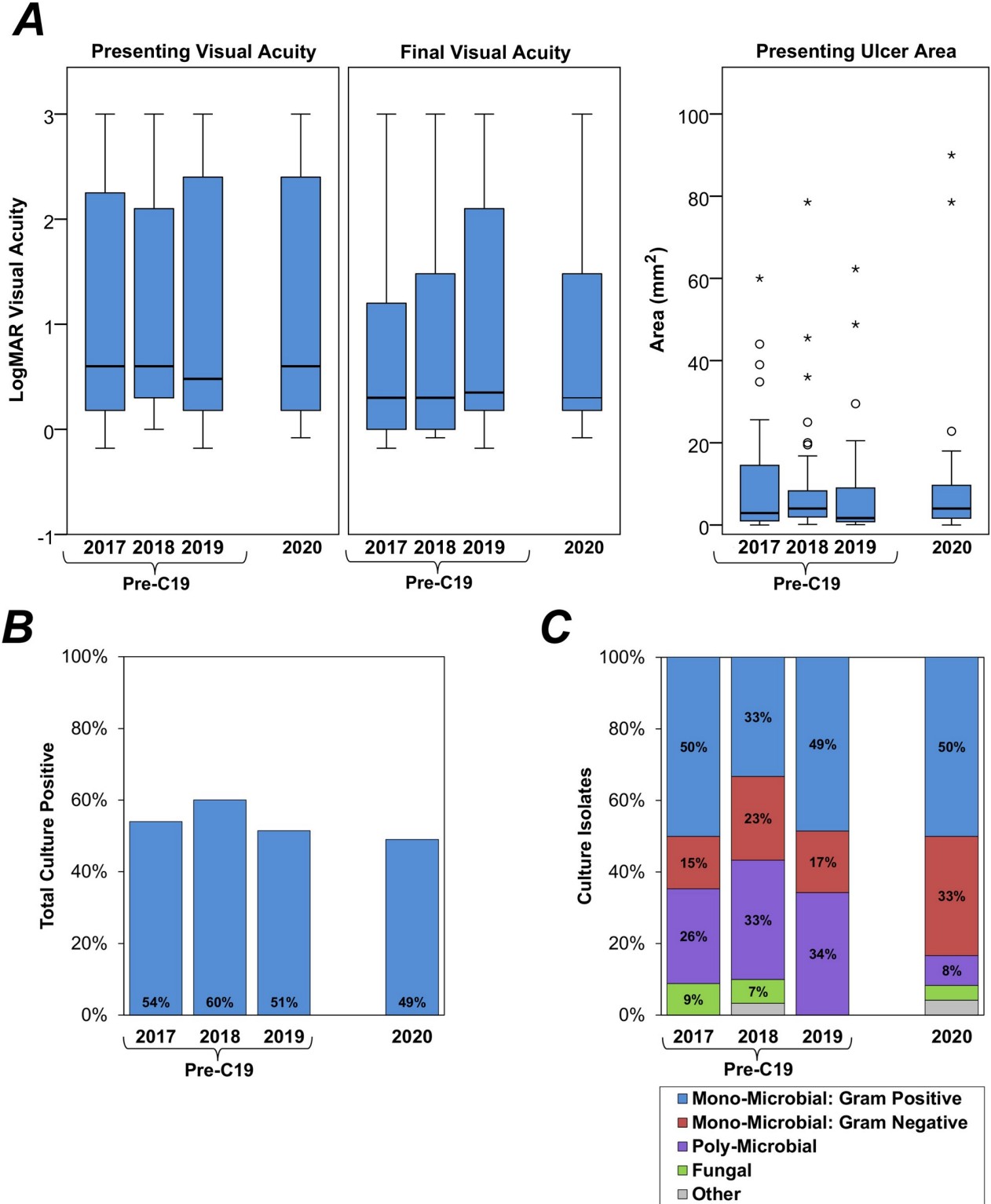

**Fig 1. Clinical assessments and microbiology by year of admission.** Clinical assessments (Fig 1A) are summarised using boxplots, with outliers indicated with circles or asterisks for those outside the box by 1.5- or 3-times the interquartile range, respectively. Microbiology is summarised as the total proportion of the cohort that were culture positive (Fig 1B), and the distribution of culture isolates from these positive cases (Fig 1C); unlabelled bars consist of <5% of cases. Further details of the definitions used for the microbial cultures are reported in Table 4.

**Table 2. Clinical assessments and outcomes.**

| | Pre-C19 | | Y2020 | | |
|---|---|---|---|---|---|
| | *N* | *Statistic* | *N* | *Statistic* | **p-Value** |
| **All Patients (N = 230)** | | | | | |
| Visual acuity at presentation (LogMAR) | 181 | 0.60 (0.18–2.10) | 49 | 0.60 (0.18–2.40) | 0.785 |
| Final visual acuity (LogMAR) | 181 | 0.30 (0.00–1.78) | 49 | 0.30 (0.18–1.48) | 0.990 |
| Ulcer area at presentation (mm$^2$) | 112 | 3.0 (1.0–9.5) | 36 | 4.0 (1.7–9.7) | 0.520 |
| Corneal involvement score | 72 | 2 (1–5) | 29 | 5 (2–5) | 0.120 |
| **Excluding patients with pre-existing visual impairment in affected eye (N = 163)** | | | | | |
| Visual acuity at presentation (LogMAR) | 126 | 0.30 (0.18–1.00) | 37 | 0.48 (0.18–1.00) | 0.608 |
| Final visual acuity (LogMAR) | 126 | 0.18 (0.00–0.60) | 37 | 0.18 (0.10–0.78) | 0.458 |
| Ulcer area at presentation (mm$^2$) | 78 | 2.0 (1.0–6.3) | 27 | 2.9 (0.8–12.0) | 0.575 |
| Corneal involvement score | 54 | 2 (1–5) | 23 | 5 (2–5) | 0.167 |

Abbreviations: Pre-C19, Pre-COVID-19 Years (2017,2018,2019); Y2020, year 2020. Data are reported as median (interquartile range), with p-values from Mann-Whitney U tests. Bold p-values are significant at $p<0.05$.

**Table 3. Clinical sequalae.**

| | Pre-C19 | Y2020 | p-Value |
|---|---|---|---|
| Admissions | 59 (32.6%) | 4 (8.2%) | **<0.001** |
| **Complications** | | | |
| Re-admission* | 1/59 (1.7%) | 0/4 (0.0%) | 1.000 |
| Corneal perforation | 12 (6.6%) | 2 (4.1%) | 0.740 |
| Endophthalmitis | 2 (1.1%) | 1 (2.0%) | 0.514 |
| Phthisis | 1 (0.6%) | 1 (2.0%) | 0.381 |
| Other complication** | 3 (1.7%) | 1 (2.0%) | 1.000 |
| **Interventions** | | | |
| Therapeutic lens | 11 (6.1%) | 0 (0.0%) | 0.126 |
| Corneal biopsy | 0 (0.0%) | 1 (2.0%) | 0.213 |
| Botulinum toxin ptosis | 2 (1.1%) | 1 (2.0%) | 0.514 |
| Corneal gluing | 11 (6.1%) | 1 (2.0%) | 0.469 |
| Evisceration | 1 (0.6%) | 0 (0.0%) | 1.000 |
| Repeat scrape | 5 (2.8%) | 2 (4.1%) | 0.643 |
| Tectonic corneal transplant | 2 (1.1%) | 0 (0.0%) | 1.000 |
| Temporary surgical tarsorrhaphy | 0 (0.0%) | 1 (2.0%) | 0.231 |
| Amniotic membrane graft | 2 (1.1%) | 0 (0.0%) | 1.000 |
| Other intervention*** | 2 (1.1%) | 3 (6.1%) | 0.066 |

Abbreviations: Pre-C19, Pre-COVID-19 Years (2017,2018,2019); Y2020, year 2020. All analyses are based on N = 181/N = 49 in the two groups, unless stated otherwise, with p-values from Fisher's exact tests. Bold p-values are significant at $p<0.05$.

*In the subgroup of patients who were admitted on initial presentation.

**Consisted of one retinal detachment, one case of corneal graft failure and one iatrogenic corneal perforation at the time of corneal scraping in the pre-C19 cohort, and a retinal detachment in the 2020 cohort.

***Consisted of one count of suture removal and one count of anterior chamber reformation following persistent aqueous humour leakage in the pre-C19 cohort, and two counts of corneal suture removal and one retinal detachment surgery in the Y2020 cohort.

**Table 4. Microbiology results summary.**

|  | **Pre-C19** | **Y2020** | **p-Value** |
|---|---|---|---|
| Total culture positive | 99/181 (54.7%) | 24/49 (49.0%) | 0.520 |
| **Culture isolates** | | | |
| Mono-microbial—gram positive | 44/99 (44.4%) | 12/24 (50.0%) | 0.654 |
| Mono-microbial—gram negative | 18/99 (18.2%) | 8/24 (33.3%) | 0.160 |
| Poly-microbial (bacterial only) | 31/99 (31.3%) | 2/24 (8.3%) | **0.022** |
| Fungal | 5/99 (5.1%) | 1/24 (4.2%) | 1.000 |
| Other* | 1/99 (1.0%) | 1/24 (4.2%) | 0.353 |
| **Most frequent gram-positive isolates**** | | | |
| Cutibacterium acnes | 19/99 (19.2%) | 1/24 (4.2%) | 0.119 |
| Staphylococcus epidermidis | 18/99 (18.2%) | 2/24 (8.3%) | 0.359 |
| Staphylococcus aureus | 11/99 (11.1%) | 6/24 (25.0%) | 0.099 |
| Streptococcus pneumoniae | 10/99 (10.1%) | 0/24 (0.0%) | 0.207 |
| **Most frequent gram-negative isolates**** | | | |
| All Moraxella species | 17/99 (17.2%) | 6/24 (25.0%) | 0.389 |
| Pseudomonas Aeruginosa | 11/99 (11.1%) | 0/24 (0.0%) | 0.120 |
| All Serratia species | 5/99 (5.1%) | 3/24 (12.5%) | 0.187 |
| Haemophilus influenzae | 2/99 (2.0%) | 1/24 (4.2%) | 0.482 |

Abbreviations: Pre-C19, Pre-COVID-19 Years (2017,2018,2019); Y2020, year 2020. p-Values are from Fisher's exact tests and bold p-values are significant at p<0.05. Bacterial isolates are presented as the number of cases in which they were isolated, and as a percentage of the total number of culture positive cases in the respective group. Poly-microbial bacterial infections were defined as more than one pathogenic bacterial isolate.

*Other cultures consisted of a mixed parasitic-bacterial case in the pre-C19 group, and a mixed fungal-bacterial infection in the Y2020 group.

**Most frequent isolates are calculated as the total frequency of isolation of each species in all the samples, accounting for both mono- and poly-microbial isolates.

MK. Although no specific restrictions were placed on ward admissions, the decrease in 2020 is likely influenced by clinicians' concerns about their patients being exposed to COVID-19 in hospital. It is interesting to note that this did not appear to have a significant impact on the measured outcomes in these patients. Before the COVID-19 pandemic, severe MK ulcers (e.g., ulcer >3mm in diameter) were admitted to the ward for intensive topical medication. However, the adjustment to self-administration of drops in 2020 appears safe and effective. The economic burden of managing MK as an in-patient is considerable [14]. While other factors (e.g. social care) may drive the need for hospital admission, judiciously increased outpatient management would help to reduce the risk of COVID-19 exposure and be significantly more cost-efficient. In this case, an estimated £150,000 of direct patient cost-savings were made in the 2020 study period [14].

Cultured isolates identified as contaminants by the microbiology department were excluded from analysis in this study; however, it can be challenging to discern contaminants from pathogenic isolates, due to the high prevalence of commensal bacteria known to cause MK [15, 25–27]. Corneal sample culture contamination may be influenced by face-mask wear. In their interesting study, Samarawickrama *et al.* [28] demonstrated the impact of study participants speaking out-loud for 30 seconds at 30cm from an open culture dish, with and without wearing a surgical mask. They found a significantly higher culture rate in the no-mask group. However, as acknowledged by the authors, their simulation likely over-estimates contamination rates compared to real-world practice, as culture plates are unlikely to have such prolonged

direct exposure, and the scrape needle (or knife) surface area is considerably smaller. Although this may explain the decrease of some oral-cavity commensals such as *Streptococcus*, it is not supported by the results of the present study when considering the prevalence of others isolates like *Staphylococcus epidermis* and the increase in *Staphylococcus aureus*. Furthermore, if mask wear reduced contamination, it would be expected that the culture positive rate in Y2020 would have reduced, relative to the earlier period. This did not occur in the present study, with the pre-C19 and Y2020 rates being similar, and comparable to other UK studies [15, 25–27].

Poor hand hygiene is a known risk factor for developing MK [29]. The bacterial diversity of the hands is greater and more dynamic than other body areas, and is considerably influenced by factors such as sex, environment and hand washing [30]. Following handwashing, although the bacterial load is decreased, its diversity is retained [30]. Since bacteria have innately varying transmission potentials, hand washing may have a differential effect on the prevention of transmission of different species [31]. Thus, the microbiological findings of this study may be influenced by the increased handwashing and the impact of campaigns advising against excessive hand-face contact during the pandemic [32], which may have altered the autoinoculation of pathogenic microbes onto the ocular surface.

The urgency with which individuals seek medical attention may differ considerably between pathologies, and delays in presentation for some conditions may be more clinically significant than for others. Mild ocular surface disease symptoms are considered by the pubic to be of low seriousness, and to require medical attention less urgently during lockdown, compared to normal circumstances [17]. Ocular trauma, specifically occurring at home, has increased during the lockdown, with delays in medical review also being reported [23, 33]. This is reflected in the higher prevalence of concurrent trauma in patients from the Y2020 group. Although uncomplicated mild trauma is relatively easily managed with lubricants and topical antibiotic prophylaxis, delays in initiating treatment may permit progression to infective keratitis. An explanation for the increased prevalence of systemic immunosuppression as a risk factor among the Y2020 cohort is less apparent.

The strength of this study is in its use of real-world data over four years in a large unit serving an out-of-hours population of up to 3.5million (5.25% of the UK population), where 72% of emergency room referrals are out of the local catchment area. This has helped to generate well-documented cohort of patients, thus reducing any variability introduced by dissimilar geography and clinical practice at different departments that may confound results. In this study, presenting VA, ulcer area, CIS, and final VA were utilised as proxy measures of severity. This represents an inherent limitation within the study; although severity scales for MK have been proposed [34, 35], there is currently no widely accepted severity stratification system that adequately covers the entire spectrum of the disease. Complications and interventions are indicators of severity; however, these relatively rare events occurred too infrequently to offer insight here. Further assessments of disease state, including a detailed time-course of lesion morphology, assessment of final optical state (corneal scarring), and patient-reported outcomes, were desirable, but not possible here. Further limitations of this study include its external generalisability, since these observations are of one centre, hence further work from multiple centres across the UK is required to validate these findings.

This study compared the features of MK patients from the first UK lockdown to previous years. These results demonstrate the considerable similarity in the presenting severity and clinical outcomes of the two groups, despite fewer patients being admitted for care in 2020. This finding is significant, considering the persisting need to safely adapt clinical practice to manage the risk of COVID-19 transmission. While other literature supports the link between increased ocular trauma and lockdown-related lifestyle changes, an explanation for the microbiology findings is less readily identifiable. Increased handwashing practices, as well as changes in

environmental factors, such as reduced contact lens wear, may have contributed to this. However, these findings must be validated on a larger scale; therefore, future work aims to connect the corneal clinician network in the UK to investigate this nationally.

## Supporting information

**S1 Table. Birmingham & Midland Eye Centre ward activity in 2015–2016.**
(DOCX)

**S2 Table. Complete list of ocular surface disease risk factors.**
(DOCX)

**S3 Table. Trends in patient characteristics across the pre-C19 period.**
(DOCX)

**S4 Table. Causes of concurrent ocular trauma.**
(DOCX)

## Acknowledgments

Dr Jonathan Swindells, Consultant Microbiologist, Black Country Pathology Services Supporting Sandwell and West Birmingham NHS Trust, for the provision of microbial keratitis culture datasets.

## Author Contributions

**Conceptualization:** Gibran F. Butt, Graham R. Wallace, Philip I. Murray, Saaeha Rauz.

**Data curation:** Gibran F. Butt, Alberto Recchioni, George Moussa.

**Formal analysis:** Gibran F. Butt, James Hodson.

**Investigation:** Gibran F. Butt.

**Methodology:** Gibran F. Butt, James Hodson.

**Project administration:** Gibran F. Butt.

**Supervision:** Philip I. Murray, Saaeha Rauz.

**Visualization:** Gibran F. Butt.

**Writing – original draft:** Gibran F. Butt, James Hodson.

**Writing – review & editing:** Gibran F. Butt, Alberto Recchioni, George Moussa, James Hodson, Graham R. Wallace, Philip I. Murray, Saaeha Rauz.

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
