## [Decision Letter · Decision Letter 0]

23 May 2021

PONE-D-21-12387

The Impact of the COVID-19 pandemic on microbial keratitis presentation patterns

PLOS ONE

Dear Dr. Rauz,

Thank you for submitting your manuscript to PLOS ONE. After careful consideration, we feel that it has merit but does not fully meet PLOS ONE’s publication criteria as it currently stands. Therefore, we invite you to submit a revised version of the manuscript that addresses the points raised during the review process.

We look forward to receiving your revised manuscript.

Kind regards,

Michael Mimouni

Academic Editor

PLOS ONE

Journal Requirements:

3a) If there are ethical or legal restrictions on sharing a de-identified data set, please explain them in detail (e.g., data contain potentially identifying or sensitive patient information) and who has imposed them (e.g., an ethics committee). Please also provide contact information for a data access committee, ethics committee, or other institutional body to which data requests may be sent.

3b) If there are no restrictions, please upload the minimal anonymized data set necessary to replicate your study findings as either Supporting Information files or to a stable, public repository and provide us with the relevant URLs, DOIs, or accession numbers. Please see http://www.bmj.com/content/340/bmj.c181.long for guidelines on how to de-identify and prepare clinical data for publication. For a list of acceptable repositories, please see http://journals.plos.org/plosone/s/data-availability#loc-recommended-repositories.

4 .Please review your reference list to ensure that it is complete and correct. If you have cited papers that have been retracted, please include the rationale for doing so in the manuscript text, or remove these references and replace them with relevant current references. Any changes to the reference list should be mentioned in the rebuttal letter that accompanies your revised manuscript. If you need to cite a retracted article, indicate the article’s retracted status in the References list and also include a citation and full reference for the retraction notice.

Reviewers' comments:

Reviewer's Responses to Questions

**Comments to the Author**

1. Is the manuscript technically sound, and do the data support the conclusions?

Reviewer #1: Yes

Reviewer #2: Yes

Reviewer #3: Yes

2. Has the statistical analysis been performed appropriately and rigorously? 

Reviewer #1: Yes

Reviewer #2: I Don't Know

Reviewer #3: I Don't Know

3. Have the authors made all data underlying the findings in their manuscript fully available?

Reviewer #1: Yes

Reviewer #2: Yes

Reviewer #3: Yes

4. Is the manuscript presented in an intelligible fashion and written in standard English?

Reviewer #1: Yes

Reviewer #2: Yes

Reviewer #3: Yes

5. Review Comments to the Author

Reviewer #1: Interesting, well written paper, valuable findings regarding COVID pandemic impact in corneal Ulcer patients. ALthough results were not what authors anticipated they kept the analysis acording to the data vailable.

Reviewer #2: I read with great interest the manuscript “ The Impact of the COVID-19 pandemic on microbial keratitis” by Gibran Farook Butt et al.

Here are some remarks that may require your attention:

1. In Methods line 112, you have stated that admission was undertaken following local guidelines- Do you have specific criteria for MK admission? If so, please elaborate on them in this section.

2. In Methods line 120, you have stated that corneal scrapes were taken if MK was suspected on presentation – do any of those patients receive antibiotics before the scraping?

3. In Methods line 145, you categorized the underlying risk factors for MK, please consider including eyelid misdirection, and exposure as a relevant risk factor for MK.

4. In Results line 189, you have mentioned the total number of MK in each year. It is interesting to evaluate the proportion of MK from total ER presented patients each year.

5. In Discussion line 286, you have postulated that patients seek healthcare for their eye symptoms less urgently in the COVID-19 era. Were more patients referred from an ophthalmologist in the community compared to self-referral? Please discuss community healthcare availability during the COVID-19 and patients' preferences to seek help in hospitals vs in first healthcare provider.

Reviewer #3: Hello,

I would like to thank the authors for presenting a nice study on microbial keratitis in the era of the COVID19 pandemic. Nonetheless, I would like to give some remarks regarding the paper:

The study does not conform with STROBE guidelines (For more information about STROBE guidelines visit https://www.strobe-statement.org/index.php?id=strobe-home):

1. A clear specified hypothesis is missing.

2. In the discussion, a ‘limitations’ paragraph is missing.

I have further remarks that are organized according to line numbers:

Line 76: 25% of ED admissions are MK

--I could not find any reference of this number in the referenced article

Line 82, 286, 341: Butt, GF, ARVO-Abstract[3546115], 2021

--Please insert a valid reference

Line 162-163

--I would specify in a clear manner that “ulcer size” is the summation of all single areas of involvement in the cornea. Similarly to the HEDS study that is referenced.

Line 166: Reference to the article of Ong et al.[20]

--I did not find any connection between the referenced article and your methodology of corneal involvement score of 0-9 points

Line 175: Statistical methods

--Why didn’t you use Chi-square test for nominal variables? Why did you choose Mann-Whitney test over student T-Test?

Line 255: ‘Rates of culture positivity rates’

--Please rephrase the sentence

Line 296-298: Poyser at al.,[10] report a decrease in contact lens associated keratitis of more than 50%, compared to the same period in 2019.

--Similarly to Agrawal Poyser et al. reports a statistically significant increase in the proportion of keratitis cases of the total emergency department admissions. It is true that the absolute number of CL associated keratitis is around half (SP1 vs. SP2) but there was no statistically significant difference in the proportions of the CL associated keratitis between the two time periods. The overall meaning of the sentence could mislead the readers.

Line 334: ‘Awareness of excessive hand to face contact’

--This fact is not mentioned in the reference article [30]

Line 394:

--I believe the full reference link should be https://www.aomrc.org.uk/wp-content/uploads/2020/04/200407_patient_public_seek_medical_help_statement.pdf

Line 413: Link for reference 6 is https://www.ctsu.ox.ac.uk/ and is non contributary

--Change the link to https://pubmed.ncbi.nlm.nih.gov/32679111/

Line 422: Link to reference 9 is /pmc/articles/PMC7350441/?report=abstract

--Change the link to https://www.ncbi.nlm.nih.gov/pmc/articles/PMC7350441/

Line 443: Link to reference 15

--https://pubmed.ncbi.nlm.nih.gov/19502241/

Line 454: Link to reference 18 is https://www.scopus.com/inward/record.uri?eid=2-s2.00025892229&doi=10.3109%2F02713689109020365&partnerID=40&md5=33c96762fb30355b 777d0cf33ff51c8c

--Correct link address should be: https://pubmed.ncbi.nlm.nih.gov/1864086/

Line 459: Link to reference 19

--https://pubmed.ncbi.nlm.nih.gov/7997324/

Line 466: Reference link address

--I suggest a pubmed link address for consistency with other addresses.https://pubmed.ncbi.nlm.nih.gov/32740065/

Line 469: Reference link address

--I suggest a pubmed link address for consistency with other addresses. https://pubmed.ncbi.nlm.nih.gov/33120625/

Line 481: Link to reference 25

--https://pubmed.ncbi.nlm.nih.gov/28452995/

Line 486: Link to reference 26

--https://pubmed.ncbi.nlm.nih.gov/29354701/

Line 488: Link to reference 27

--https://pubmed.ncbi.nlm.nih.gov/28813424/

Line 498: Link to reference 30

--https://pubmed.ncbi.nlm.nih.gov/32839091/

Line 499-510: Links to references

--Correct to valid and consistent pubmed links

Sincerely yours,

NS

6. PLOS authors have the option to publish the peer review history of their article (what does this mean?). If published, this will include your full peer review and any attached files.

Reviewer #1: No

Reviewer #2: **Yes: **Efrat Naaman

Reviewer #3: No

---

## [Author Response · Author response to Decision Letter 0]

9 Jul 2021

Dear PLOSONE editorial team,

Many thanks for considering our paper for publication at your prestigious journal. We are grateful for the reviewers’ time and insights which have helped to improve the paper. Below are the reviewers’ comments (in black) and the authors’ responses (red). 

1- Style updated as per requirements

Thank you for directing us to the journal style guidance. The manuscript has been updated accordingly. 

2- Details regarding consent updated in manuscript with the below statement

This retrospective study of medical records was conducted in accordance with the Declaration of Helsinki and in accordance with local institutional policy. Utilising the National Health Service Health Research Authority & UKRI decision tool this study is not research as defined by the UK Policy Framework for Health and Social Care Research (see attached pdf “MK–C19_Result-NOT Research”). All data were anonymised prior to analysis and the need for consent was waived. Approval was obtained from Sandwell and West Birmingham Hospitals NHS Trust Department of Clinical Effectiveness (registration #1512) to undertake this project as a service evaluation. 

3- Data sharing ? 

Thank you for highlighting this important aspect of open research. The study data have been uploaded to Figshare - DOI https://doi.org/10.6084/m9.figshare.14873052

Reviewer 2 comments

1. In Methods line 112, you have stated that admission was undertaken following local guidelines- Do you have specific criteria for MK admission? If so, please elaborate on them in this section.

We thank the reviewer for prompting this clarification. The decisions to admit are based upon a number of factors listed in the local guidelines and are taken under consideration on case-by-case basis. We have clarified these factors in the manuscript – Lines 131-135:

“The decision to admit patients was based on factors including clinical severity, risk of adverse events (e.g., perforation), social care needs (e.g., the ability to diligently administer all drops day and night, residential proximity to healthcare premises and ability to attend for daily visits), with the final decision being taken by the lead clinician for any given session. “

2. In Methods line 120, you have stated that corneal scrapes were taken if MK was suspected on presentation – do any of those patients receive antibiotics before the scraping?

Typically a scrape will be taken where MK is suspected on presentation. Where a re-scrape is indicated (e.g., poor response to treatment) then antibiotics are withheld for 24 hours before sampling. Patients may be using long term antibiotics (topical or systemic) for other indications or they may be initiated prior to presenting at medical services (e.g., self-medicated obtained over the counter, seen previously at other healthcare services sites (primary or secondary care, independent healthcare provider etc.). For inclusion into this study, patients were required to have undergone a corneal scrape – no restrictions were applied regarding prior medication. The authors acknowledge that it would interesting to understand the impact of pre-existing antibiotic use on the final outcome however within our sample the numbers of such patients were too low to for sub-group analyses [2017 (n =9), 2018 (n=4), 2019 (n= 7), 2020 (n= 12)].

3. In Methods line 145, you categorized the underlying risk factors for MK, please consider including eyelid misdirection, and exposure as a relevant risk factor for MK.

Many thanks to the reviewer for bringing our attention to the importance of a detailed explanation of the associated risk factors. Among others, the suggested risk factor is included in our classification. The complete list of risk factors taken into account in this study is listed in supplementary table 2.

4. In Results line 189, you have mentioned the total number of MK in each year. It is interesting to evaluate the proportion of MK from total ER presented patients each year.

Thank you for these interesting suggestions. We have updated the manuscript to comment on this, as below. 

Lines 206-209

“Total numbers of attendances to the BMEC for any indication were 12,128 during the time window in 2018 and 12,239 in 2019, compared to only 5,759 in 2020 (data were not available for 2017). As such, MK comprised 0.5% of attendances in 2018-19, which increased significantly to 0.9% in 2020 (p=0.001). During period being studied in 2020, all first-time face to face appointments were replaced with an initial telephone consultation, followed by a face-to-face consultation, where indicated. This had the effect of reducing the overall number of face-to-face appointments [22].”

5. In Discussion line 286, you have postulated that patients seek healthcare for their eye symptoms less urgently in the COVID-19 era. Were more patients referred from an ophthalmologist in the community compared to self-referral? Please discuss community healthcare availability during the COVID-19 and patients' preferences to seek help in hospitals vs in first healthcare provider.

We thank the reviewer for raising this interesting discussion point. Within our region, community eye care is led by general practitioners together with community independent optometry services. This is a feature of the eye care system in England. Since both of these services were operating at severely reduced capacity (optometry services were completely closed for a period of time), patients were presenting to hospital eye services through alternate pathways (e.g. the NHS phoneline 111). Crucially, patients with microbial keratitis would not be treated independently by either of these services, and would present or be referred to hospital services in any event. Therefore, the number of presentations of MK to our unit would be unchanged., tThese numbers are, therefore, a representation of the regional incidence of MK. We have clarified this within the manuscript as below, lines 79-84.

“Typically, patients with MK would be managed by hospital services, presenting either by self-referral, or by referral from community-based eye care services. Such services are typically delivered by community opticians and general practitioners. During the pandemic, community healthcare services were severely impaired, with more patients instead utilising alternate pathways, such as the NHS phoneline 111.” 

 

Reviewer 3 comments

Reviewer #3: Hello,

I would like to thank the authors for presenting a nice study on microbial keratitis in the era of the COVID19 pandemic. Nonetheless, I would like to give some remarks regarding the paper:

The study does not conform with STROBE guidelines (For more information about STROBE guidelines visit https://www.strobe-statement.org/index.php?id=strobe-home):

1. A clear specified hypothesis is missing.

This has been added into the manuscript, lines 95-96.

“We hypothesized that the presentation and outcomes of patients with MK would be affected by the first UK national lockdown in 2020, compared to previous years.”

2. In the discussion, a ‘limitations’ paragraph is missing.

The authors thank the reviewer for highlighting the importance of a limitations section. This is included in the manuscript as the penultimate paragraph in the discussion (line 374-388). It has been amended to emphasise this as below, lines 374-388.

“The strength of this study is in its use of real-world data over four years in a large unit serving an out-of-hours population of up to 3.5million (5.25% of the UK population) where 72% of emergency room referrals are out of the local catchment area. This has helped to generate well-documented cohorts of patients, thus reducing any variability introduced by dissimilar geography and clinical practice at different departments that may confound results. In this study, presenting VA, ulcer area, CIS, and final VA were utilised as proxy measures of severity. This represents an inherent limitation within the study, although severity scales for MK have been proposed [32,33], there is currently no widely accepted severity stratification system that adequately covers the entire spectrum of the disease. Complications and interventions are indicators of severity; however, these relatively rare events occurred too infrequently to offer insight here. Further assessments of disease state, including a detailed time-course of lesion morphology, assessment of final optical state (corneal scarring), and patient-reported outcomes, were desirable, but not possible here. Further limitations of this study include its external generalisability, since these observations are of one centre, further work from multiple centres across the UK is required to confirm these findings.”

I have further remarks that are organized according to line numbers:

Line 76: 25% of ED admissions are MK

--I could not find any reference of this number in the referenced article

The authors thank the reviewer for prompting clarification of this point. This statement is clarified in the manuscript with the addition of supplementary information (Supplementary Table 1) that demonstrates the annual emergency ward admission rate to be approximately 450 cases, and the annual number of microbial keratitis admission to be approximately 100. Therefore, making microbial keratitis approximately 25% of the emergency admission to our unit. 

Line 82, 286, 341: Butt, GF, ARVO-Abstract[3546115], 2021

--Please insert a valid reference

Thanks to the reviewer for directing the authors to correct this. This has been updated.

Line 162-163

--I would specify in a clear manner that “ulcer size” is the summation of all single areas of involvement in the cornea. Similarly to the HEDS study that is referenced.

This has been clarified as suggested. Line 178 -181

“The area was then calculated by multiplying these readings together; epithelial defect, infiltrate, and scar size were not differentiated, henceforth this measurement is referred to as the “ulcer area”, which was also the summation of all single areas of involvement in the cornea.”

Line 166: Reference to the article of Ong et al.[20]

--I did not find any connection between the referenced article and your methodology of corneal involvement score of 0-9 points

The Ong et al study details the derivation of a corneal opacification and vascularisation score using a scoring system based on the involved regions of the cornea. The tool is validated in the reference, and is available to download as follows:

• → Ong et al Paper: https://www.sciencedirect.com/science/article/pii/S1542012419302101

• →Appendix A: Supplementary data 

• →Cited as Supplementary Appendix 2 

It describes a method for the corneal involvement score (opacity and vascularisation). The present study describes the site of the abscess derived of a corneal involvement score using a similar methodology as the Ong paper, which is validated and peer reviewed, hence it is referenced for further details. Below is an excerpt from the Ong et al paper explaining their methodology. 

“Corneal vascularisation and corneal opacity: assess on the slit lamp. The central cornea is defined as the central 4 mm of the cornea as delineated by a slit‐beam scale. Each peripheral corneal quadrant is scored as positive with a score of 1 for involvement by vessels or opacity separately. For corneal vascularisation, a score of 1 is given to each peripheral quadrant involved; the central cornea, if involved, is scored as 1 giving a maximum ocular surface vascularisation score of 5. For corneal opacification, a score of 1 is given to each peripheral quadrant involved and an additional score of 5 if the central cornea is involved, giving a maximum score of 9.”

Line 175: Statistical methods

--Why didn’t you use Chi-square test for nominal variables? Why did you choose Mann-Whitney test over student T-Test?

Fisher’s exact tests were selected because it is an exact test (as opposed to the approximate estimate provided by the chi squared test) and also because it is the preferred test for lower sample sizes, where chi squared should be avoided. 

The Mann-Whitney U test is a nonparametric test that allows two groups or conditions or treatments to be compared without making the assumption that values are normally distributed. In the present study the Mann-Whitney U tests were used, as the majority of continuous variables were not normally distributed.

Line 255: ‘Rates of culture positivity rates’

--Please rephrase the sentence

The authors thank the reviewer for prompting this clarification. It has been amended as suggested, line 276.

Line 296-298: Poyser at al.,[10] report a decrease in contact lens associated keratitis of more than 50%, compared to the same period in 2019.

--Similarly to Agrawal Poyser et al. reports a statistically significant increase in the proportion of keratitis cases of the total emergency department admissions. It is true that the absolute number of CL associated keratitis is around half (SP1 vs. SP2) but there was no statistically significant difference in the proportions of the CL associated keratitis between the two time periods. The overall meaning of the sentence could mislead the readers.

Many thanks for highlighting the importance of clarity in this section of the discussion. The sentence clarified as below, line 316-321. 

“Disease epidemiology and health care services vary geographically. Whilst Agarwal et al. [22] reported an increase in MK incidence during the lockdown at their unit in India, Poyser at al.,[10] report a decrease in contact lens associated keratitis of more than 50%, compared to the same period in 2019, although the proportions remained similar in both study periods. The present study’s results identified an increase in the proportion of MK patients seen in the department compared to previous years, whilst fewer patients were seen in the department overall [23].”

Line 334: ‘Awareness of excessive hand to face contact’

--This fact is not mentioned in the reference article [30]

The reference and text have been updated in the manuscript as follows, line 358-361.

“Thus, the microbiological findings of this study may be influenced by the increased handwashing and the impact of campaigns advising against excessive hand-face contact during the pandemic [29], which may have altered the autoinoculation of pathogenic microbes onto the ocular surface.”

Line 394:

--I believe the full reference link should be https://www.aomrc.org.uk/wp-content/uploads/2020/04/200407_patient_public_seek_medical_help_statement.pdf

Line 413: Link for reference 6 is https://www.ctsu.ox.ac.uk/ and is non contributary

--Change the link to https://pubmed.ncbi.nlm.nih.gov/32679111/

Line 422: Link to reference 9 is /pmc/articles/PMC7350441/?report=abstract

--Change the link to https://www.ncbi.nlm.nih.gov/pmc/articles/PMC7350441/

Line 443: Link to reference 15

--https://pubmed.ncbi.nlm.nih.gov/19502241/

Line 454: Link to reference 18 is https://www.scopus.com/inward/record.uri?eid=2-s2.00025892229&doi=10.3109%2F02713689109020365&partnerID=40&md5=33c96762fb30355b 777d0cf33ff51c8c

--Correct link address should be: https://pubmed.ncbi.nlm.nih.gov/1864086/

Line 459: Link to reference 19

--https://pubmed.ncbi.nlm.nih.gov/7997324/

Line 466: Reference link address

--I suggest a pubmed link address for consistency with other addresses.https://pubmed.ncbi.nlm.nih.gov/32740065/

Line 469: Reference link address

--I suggest a pubmed link address for consistency with other addresses. https://pubmed.ncbi.nlm.nih.gov/33120625/

Line 481: Link to reference 25

--https://pubmed.ncbi.nlm.nih.gov/28452995/

Line 486: Link to reference 26

--https://pubmed.ncbi.nlm.nih.gov/29354701/

Line 488: Link to reference 27

--https://pubmed.ncbi.nlm.nih.gov/28813424/

Line 498: Link to reference 30

--https://pubmed.ncbi.nlm.nih.gov/32839091/

Line 499-510: Links to references

--Correct to valid and consistent pubmed links

Many thanks indeed to the reviewer for their detailed comments regarding these references and for signposting to the appropriate URLS. All references have been updated to include the DOIs. 

Yours sincerely, 

Gibran F Butt (First author) - g.f.butt@bham.ac.uk

Alberto Recchioni

George Moussa

James Hodson

Graham R Wallace

Philip I Murray 

Saaeha Rauz (Corresponding and senior author) - s.rauz@bham.ac.uk

---

## [Decision Letter · Decision Letter 1]

3 Aug 2021

The Impact of the COVID-19 pandemic on microbial keratitis presentation patterns

PONE-D-21-12387R1

Dear Dr. Rauz,

We’re pleased to inform you that your manuscript has been judged scientifically suitable for publication and will be formally accepted for publication once it meets all outstanding technical requirements.

Kind regards,

Michael Mimouni

Academic Editor

PLOS ONE

Additional Editor Comments (optional):

Reviewers' comments:

Reviewer's Responses to Questions

**Comments to the Author**

1. If the authors have adequately addressed your comments raised in a previous round of review and you feel that this manuscript is now acceptable for publication, you may indicate that here to bypass the “Comments to the Author” section, enter your conflict of interest statement in the “Confidential to Editor” section, and submit your "Accept" recommendation.

Reviewer #2: All comments have been addressed

Reviewer #3: All comments have been addressed

2. Is the manuscript technically sound, and do the data support the conclusions?

Reviewer #2: (No Response)

Reviewer #3: Yes

3. Has the statistical analysis been performed appropriately and rigorously? 

Reviewer #2: (No Response)

Reviewer #3: Yes

4. Have the authors made all data underlying the findings in their manuscript fully available?

Reviewer #2: (No Response)

Reviewer #3: Yes

5. Is the manuscript presented in an intelligible fashion and written in standard English?

Reviewer #2: (No Response)

Reviewer #3: Yes

6. Review Comments to the Author

Reviewer #2: (No Response)

Reviewer #3: I would like to thank the authors for addressing all of my comments and for revising the manuscript. I enjoyed reading your article and learned from it. I am sure that our readers would also find it interesting and enriching.

Sincerely,

N.S

7. PLOS authors have the option to publish the peer review history of their article (what does this mean?). If published, this will include your full peer review and any attached files.

Reviewer #2: No

Reviewer #3: **Yes: **Nir Stanescu

---

## [Editor Report · Acceptance letter]

9 Aug 2021

PONE-D-21-12387R1 

The Impact of the COVID-19 pandemic on microbial keratitis presentation patterns. 

Dear Dr. Rauz:

I'm pleased to inform you that your manuscript has been deemed suitable for publication in PLOS ONE. Congratulations! Your manuscript is now with our production department. 

Kind regards, 

on behalf of

Dr. Michael Mimouni 

Academic Editor

PLOS ONE